# Enhancement of AFB_1_ Removal Efficiency via Adsorption/Photocatalysis Synergy Using Surface-Modified Electrospun PCL-g-C_3_N_4_/CQDs Membranes

**DOI:** 10.3390/biom13030550

**Published:** 2023-03-17

**Authors:** Liangtao Yao, Changpo Sun, Hui Lin, Guisheng Li, Zichao Lian, Ruixin Song, Songlin Zhuang, Dawei Zhang

**Affiliations:** 1Engineering Research Center of Optical Instrument and System, Ministry of Education and Shanghai Key Lab of Modern Optical System, University of Shanghai for Science and Technology, No.516 Jungong Road, Shanghai 200093, China; 2Standards and Quality Center of National Food and Strategic Reserves Administration, No.25 Yuetan North Street, Xicheng District, Beijing 100834, China; 3Department of Chemistry, College of Science, University of Shanghai for Science and Technology, No.516 Jungong Road, Shanghai 200093, China; 4Fujian Provincial Key Laboratory for Advanced Micro-Nano Photonics Technology and Devices, Research Center for Photonics Technology, Quanzhou Normal University, Quanzhou 362000, China

**Keywords:** PCL electrospun membranes, aflatoxin B_1_, adsorption, photocatalysis, g-C_3_N_4_/CQDs, visible light

## Abstract

Aflatoxin B_1_ (AFB_1_) is a highly toxic mycotoxin produced by aspergillus species under specific conditions as secondary metabolites. In this study, types of PCL (Polycaprolactone) membranes anchored (or not) to g-C_3_N_4_/CQDs composites were prepared using electrospinning technology with (or without) the following surface modification treatment to remove AFB_1_. These membranes and g-C_3_N_4_/CQDs composites were characterized by SEM, TEM, UV-vis, XRD, XPS and FTIR to analyze their physical and chemical properties. Among them, the modified PCL-g-C_3_N_4_/CQDs electrospun membranes exhibited an excellent ability to degrade AFB_1_ via synergistic effects of adsorption and photocatalysis, and the degradation rate of 0.5 μg/mL AFB_1_ solution was observed to be up to 96.88% in 30 min under visible light irradiation. Moreover, the modified PCL-g-C_3_N_4_/CQDs electrospun membranes could be removed directly after the reaction process without centrifugal or magnetic separation, and the regeneration was a green approach synchronized with the reaction under visible light avoiding physical or chemical treatment. The mechanism of adsorption by electrostatic attraction and hydrogen bonding interaction was revealed and the mechanism of photodegradation of AFB_1_ was also proposed based on active species trapping experiments. This study illuminated the highly synergic adsorption and photocatalytic AFB_1_ removal efficiency without side effects from the modified PCL-g-C_3_N_4_/CQDs electrospun membranes, thereby offering a continual and green solution to AFB_1_ removal in practical application.

## 1. Introduction

At present, the biological contamination of food has been paid more and more attention, along with the issue of food safety. Mycotoxin contamination is one of the main factors causing food safety problems. There are many kinds of mycotoxins, among which aflatoxin B_1_ (AFB_1_) is the most toxic biological toxin food produced by aspergillus species so far. Its toxicity is 10 times that of potassium cyanide and 68 times that of arsenic, and it is classified as a class I human carcinogen by the International Agency for Research on Cancer (IARC) [1]. AFB_1_ can easily enter the human food chain and threaten people’s health. In order to ensure human health from the harm of AFB_1_, international organizations and countries around the world have determined the maximum tolerable limits of AFB_1_ in various foods. In the European Commission [2], the maximum limits are 2 μg/kg for AFB_1_ in edible oils, cereals and cereal products. In China [3], the maximum limits of AFB_1_ are set at 20 μg/kg for peanut and maize oils, and 10 μg/kg for the other vegetable oils. In the United States [4], the maximum limits of total aflatoxins (AFB_1_ + AFB_2_ + AFG_1_ + AFG_2_) in all foods (except milk) are 20 μg/kg; acceptable levels of AFM_1_ in milk and dairy products are 0.5 μg/kg.

Studies have shown that intake of large amounts of AFB_1_ in a short time can lead to liver damage, such as liver tissue hemorrhage and acute hepatitis. To reduce or eliminate the adverse effects caused by AFB_1_, on the one hand, it is necessary to control the growth of the aspergillus species or hold back the arise of AFB_1_ [5]; on the other hand, the symptoms of AFB1 poisoning can be effectively relieved by taking animal function regulators (curcumin) or antiaflatoxin bacteria (probiotics) [6]. Furthermore, we need to develop various effective and practical methods for the detoxification of AFB_1_ [7]. Adsorption is a widely used method to decrease aflatoxin contamination. There has been a series of studies on the elimination of AFB_1_ with adsorbents. Ma et al. applied copper-based metal-organic frameworks (MOFs) to synthesize the porous carbonaceous materials as sorbents for the removing of AFB_1_ from plant oils, and removed more than 90% of AFB_1_ within 30 min [8]. Phillips et al. synthesized a highly active sodium bentonite clay with enhanced AFB_1_ sorption efficacy compared with bentonite clay and other clays [9]. Karmanov et al. studied the performance of adsorption-desorption of aflatoxin B2 (AFB_2_) using lignins obtained from several cultivated and medicinal plants in an in vitro simulated gastrointestinal tract environment; lignins from ledum and jerusalem artichoke exhibited the highest AFB_2_ adsorption capacity, and the chemisorption mechanisms played the most leading role [10]. However, the adsorption capacity gradually decreases as the adsorbents are saturated with the adsorbed AFs. In addition, some adsorbents are short of reusability due to the shortage of environmentally friendly regeneration methods. For instance, common powder adsorbents usually need centrifuge separation for adsorbents recovery [11], and the regeneration process usually requires solvent washing under acidic/alkaline conditions or undergoing calcination treatment, which would inevitably release pernicious chemicals into the environment [12].

As we all know, adsorption membranes prepared by electrospinning technology have attracted sense of attention in wastewater treatment due to their large specific surface area [13,14]. Many kinds of pristine electrospinning polymer adsorption membranes or electrospinning polymer adsorption membranes containing additives (such as graphene oxide [15], silica [16], and carbon nanotubes [17]) were developed to adsorb organic pollutants or heavy metal ions [18,19]. These membranes were easy to separate from the reaction substrate after adsorption reaction, thus avoiding centrifugal separation and possible secondary pollution. Therefore, it is a reasonable strategy to use electrospun membranes to absorb AFB_1_ in an aqueous medium. However, these electrospun membranes still need to be treated by chemical or physical methods for regeneration. Are there recyclable membranes that can not only degrade AFB_1_ efficiently, but also avoid releasing harmful chemicals to the environment during regeneration?

Photocatalytic technology is a newly developed technology to degrade pollutants under mild or gentle conditions including AFB_1_ [20,21]. In the photocatalytic process, When the light with appropriate energy irradiates the photocatalysts, the electrons (e^−^) are excited from the valence band to the conduction band leaving behind holes (h^+^) [22]. Then, these photogenerated charges (e^−^ and h^+^) migrate from the inner of the photocatalysts to the surface of the photocatalysts. These photogenerated charges interact with H_2_O, O_2_ or OH^−^ around to produce •OH and •O_2_^−^, which can attack the pollutant molecules into smaller fragments, even CO_2_ or H_2_O [23]. Compared with traditional treatment methods, photocatalysis technology is environmentally friendly and low-cost. Inspired by the process mentioned above, it is an attractive strategy to develop a kind of green and recyclable membrane that combines adsorption and photocatalysis techniques to degrade AFB_1_ synergistically through electrospinning technology, overcoming difficulties in separating traditional adsorbents and photocatalysts.

Among the wide variety of polymers that can be used for electrospinning [24,25], polycaprolactone (PCL) has attracted extensive interest due to its availability, non-toxicity, and stability [26,27]. More importantly, the electrospun membranes prepared by PCL have excellent mechanical properties [28]. In this paper, PCL was chosen as the raw material of electrospun membranes. In order to overcome the disadvantage of poor adsorption capacity of the PCL electrospun membranes, surface modification technology was used to functionalize the membranes. Polydopamine (PDA) modification inspired by the excellent adhesion of *Mytilus edulis foot* has been widely used in membrane surface modification. Dopamine (DA) can self-polymerize to form a PDA layer coating at plenty of substrates under alkaline conditions or air atmosphere [29], which can improve the hydrophilicity and mechanical properties of the modified membranes. Furthermore, the phenolic hydroxyl groups on the surface of the PDA layer can be used to conjugate with amino-containing compounds to capture target molecules intentionally. So far, it has been reported that polyethyleneimine (PEI) [30], polyacrylamide (PAAM) [31], and other amino-containing compounds were grafted onto PDA-modified membranes to adsorb CO_2_, dye molecules, protease and other substances. In this study, we used PEI conjugate onto PDA-modified PCL electrospun membranes to adsorb AFB_1_.

In terms of photocatalysts, graphitic carbon nitride (g-C_3_N_4_), a metal-free semiconductor, has attracted wide attention due to its environmental friendliness, easy modification, and proper band gap [32]. However, the high recombination rate of photogenerated charges and poor spectral response range have become crucial issues for pristine g-C_3_N_4_ [33]. Metal/non-metal element doping [34,35], construction of heterojunction [36], and other strategies were used to optimize the photocatalytic performance of g-C_3_N_4_. Carbon quantum dots (CQDs), a “zero-dimensional” nanomaterial, has become one of the most promising co-catalysts due to their excellent electron transfer ability and up-converting photoluminescence [37]. Therefore, it is a reasonable way to modify g-C_3_N_4_ with CQDs to enhance visible light absorption as well as reduce the recombination of photogenerated charges [38]. The raw materials (such as urea, melamine, citric acid, sucrose, etc.) required for the preparation of CQDs and g-C_3_N_4_ are easy to obtain and the preparation process is simple, which is favorable for practical applications. In this paper, g-C_3_N_4_/CQDs composites were used for the photocatalytic degradation of AFB_1_ under visible light.

Herein, the flexible PCL-g-C_3_N_4_/CQDs electrospun membranes were successfully prepared using electrospinning technology. The surface of electrospun membranes were modified by PDA and PEI to remove AFB_1_ continuously. To study the synergistic effect of adsorption and photocatalysis, comparative experiments were carried out using pristine PCL electrospun membranes, modified PCL electrospun membranes, PCL-g-C_3_N_4_/CQDs electrospun membranes and modified PCL-g-C_3_N_4_/CQDs electrospun membranes. The adsorption mechanism and the photocatalytic degradation mechanism of AFB_1_ were investigated in the presence of different sacrificial agents. Moreover, the effect of recycling on the adsorption and photocatalytic efficiency was also evaluated. The work presented in this paper has potential practical value for the degradation of AFB_1_, which has plagued the food industry for a long time.

## 2. Experiment

### 2.1. Materials and Reagents

AFB_1_ was purchased from Beijing Puhuashi Technology Development Co., Ltd. (Beijing, China). and dissolved to a certain concentration with deionized water. Urea (≥99.0% purity), anhydrous citric acid (≥99.0% purity), anhydrous methanol (≥99.0% purity) polycaprolactone (PCL, M_w_ ≈ 80,000), dopamine hydrochloride (AR, 99.5%), polyethyleneimine (PEI, M_w_ ≈ 10,000), N,N-dimethylformamide (DMF, AR, 99.5%), hexafluoroisopropanol (HFIP, AR, 99.5%), sodium hydroxide (AR, 99.5%) and Tris-HCl buffer (pH = 8.5) were purchased from Jingji Co., Ltd. (Suzhou, China). Glacial acetic acid (for HPLC, ≥99.9%), methanol (for HPLC, ≥99.9%), trifluoroacetate (for HPLC, ≥99.5%) and acetonitrile (for HPLC, ≥99.9%) were purchased from Macklin Biochemical Co., Ltd. (Shanghai, China). All reagents were used without any further purification. Dialysis bags were purchased from Shanghai yuan ye Bio-Technology Co., Ltd. (Shanghai, China). The deionized water used in this work was purified using the Millipore system purchased from Merck Co., Ltd. (Shanghai, China).

### 2.2. Preparation of g-C_3_N_4_/CQDs Composites

Powdered g-C_3_N_4_ was prepared by calcining urea at 550 °C for 3 h (5 °C/min). CQDs were synthesized through a facile hydrothermal method according to the improved method in reference [39]. Briefly, a certain amount of urea and citric acid with a mass ratio of 48:19 were fully dissolved in DMF. Then the solution was transferred to a polytetrafluoroethylene reactor, followed by placing it in a drying furnace at 180 °C for 6 h. After the reaction, on cooling to room temperature, the obtained red solution was centrifuged at a speed of 8000 r·min^−1^ for 10 min to remove the large deposit, and dialysis was carried out in a dialysis bag with a molecular weight of 500 Da for 48 h. Finally, the CQDs was obtained after freeze-drying treatment.

In a typical preparation of g-C_3_N_4_/CQDs composites, a certain amount of g-C_3_N_4_ was dissolved in anhydrous methanol. Then, CQDs were added into the above solution, stirred and ultrasonicated for 1 h, respectively. After the suspension was dried at 60 °C for 6 h, g-C_3_N_4_/CQDs composites were obtained. The photocatalytic degradation performance of g-C_3_N_4_/CQDs samples with different contents of CQDs (0.1%, 0.3%, 0.5%, and 0.7%) and pristine g-C_3_N_4_ can be seen in Appendix A, we determined that the mass ratio of CQDs in g-C_3_N_4_/CQDs composites was 0.5% in this study.

### 2.3. Preparation of Modified PCL-g-C_3_N_4_/CQDs Electrospun Membranes

The modified PCL-g-C_3_N_4_/CQDs electrospun membranes were prepared by electrospinning and the following surface modification treatment. The schematic illustration was shown in Figure 1. Typically, 0.2 g g-C_3_N_4_/CQDs composites were added into 10 mL HFIP and ultrasonicated for 1h to fully disperse. Subsequently, 1 g PCL was added and stirred for 12 h at 50 °C to obtain a yellow-grey solution. Then loaded the prepared solution into a 10 mL plastic syringe with a metal needle. The syringe was driven by a hydraulic pump for electrospinning at a flow rate of 2.5 mL/h. The applied voltage during electrospinning was 20 kV and the distance from the aluminum foil surface to the metallic needle was 15 cm. After electrospinning, the PCL-g-C_3_N_4_/CQDs electrospun membranes were dried at 45 °C in a drying furnace for 12 h.

Surface modification treatment was a two-step process that combines hydrolysis reaction with subsequent grafting technology, which was briefly described as follows. First, immersed the PCL-g-C_3_N_4_/CQDs electrospun membranes in 2 g·L^−1^ DA hydrochloride solution (Tris-HCl buffer, pH = 8.5) at 45 °C with 120 r·min^−1^ agitations to form PDA layer for 12 h. Second, placed the washed PDA-coated PCL-g-C_3_N_4_/CQDs electrospun membranes by deionized water in 2 g·L^−1^ PEI aqueous solution at 45 °C with 120 r·min^−1^ agitations for 12 h. Finally, the modified PCL-g-C_3_N_4_/CQDs electrospun membranes were obtained after washing and drying treatment.

Pristine PCL electrospun membranes, modified PCL electrospun membranes and PCL-g-C_3_N_4_/CQDs electrospun membranes were prepared in a similar manner, respectively.

### 2.4. Membranes Characterization

The morphologies of the pristine PCL electrospun membranes, modified PCL electrospun membranes, PCL-g-C_3_N_4_/CQDs electrospun membranes and modified PCL-g-C_3_N_4_/CQDs electrospun membranes were observed using scanning electron microscopy (SEM, ZEISS Sigma, Germany) and the microstructures of g-C_3_N_4_/CQDs composites were observed by transmission electron microscopy (TEM, JEM-2100F). The UV-vis absorption spectra and photoluminescence of g-C_3_N_4_/CQDs composites were measured with a UV-vis-NIR double beam spectrophotometer (Lambda 1050, Perkin-Elmer, Waltham, MA, USA) and a steady-state/transient fluorescence spectrometer (Edinburgh Instruments, Livingston FL1000, UK). X-ray diffraction (XRD) patterns were obtained with an x-ray diffractometer (XRD, MiniFlex 600, Rigaku, The Woodlands, TX, USA) at a scanning speed of 2°/min. Fourier-transformed infrared (FTIR) spectra were analyzed on a fourier infrared spectrometer (Vector-22, BRUKER, wavenumber region: 500~4000 cm^−1^, wavenumber resolution: 1 cm^−1^). High-resolution XPS spectra were analyzed using an X-ray photoelectron spectrometer (NexsaG2, Thermo Scientific, Waltham, Ma, USA, voltage: 12 kV, energy range: 100~700 eV, scanning step: 0.05 eV).

### 2.5. Photocatalytic Degradation Experiment

The adsorption experiments were performed at room temperature in the dark by immersing 0.05 g of pristine PCL electrospun membranes, modified PCL electrospun membranes, PCL-g-C_3_N_4_/CQDs electrospun membranes and modified PCL-g-C_3_N_4_/CQDs electrospun membranes into 50 mL of AFB_1_ aqueous solutions (0.5 μg/mL), respectively. During the experiments, 0.5 mL of the AFB_1_ aqueous solution was taken every 5 min within 30 min.

The adsorption-photocatalysis experiments were performed under visible light radiation, whereas the other concentrations were kept constant with adsorption experiments. A 300 W xenon lamp with a 400 nm cut-off filter was used as a light source. The distance between the Xenon lamp and the AFB_1_ aqueous surface was 10 cm. During the experiments, 0.5 mL of the AFB_1_ aqueous solution was taken every 5 min for a total of 30 min.

In this study, the concentrations of the AFB_1_ were analyzed by the Waters-600 high-performance liquid chromatography (HPLC) equipped with a UV/Visible detector (emission wavelength at 365 nm) and C-18 Phenomenex reverse phase column (250 × 4.6 mm i.d., 5 μm) at a flow rate of 1 mL/min with an isocratic system composed of acetonitrile: methanol: water (10:20:70). The total running time was 20 min and the injection volume was 10 μL.

The stability of the modified PCL-g-C_3_N_4_/CQDs electrospun membranes was evaluated by 5 continuous cycle experiments. NaCl (0.1 mol) and urea (0.1 mol) were added into AFB_1_ solutions in the dark, respectively, before adsorption experiments as electrostatic and hydrogen bond inhibitors to explore the adsorption mechanism of AFB_1_. In order to understand the mechanism of photodegradation of AFB_1_ by the electrospun membranes, active species trapping experiments were carried out with the addition of ammonium oxalate (AO, 1 mM), isopropanol (IPA, 1 mM), and 1,4-benzoquinone (BQ, 1 mM) to capture photogenerated holes (h^+^), hydroxyl radicals (•OH) and super-oxide anion radicals (•O_2_^−^), respectively.

## 3. Results and Discussion

### 3.1. Micro-Structure of PCL-g-C_3_N_4_/ CQDs Electrospun Membranes

The morphologies of the pristine PCL electrospun membranes, modified PCL electrospun membranes, PCL-g-C_3_N_4_/CQDs electrospun membranes and modified PCL-g-C_3_N_4_/CQDs electrospun membranes were displayed in Figure 2. Figure 2a presented the SEM images of the pristine PCL electrospun membranes, and these smooth PCL nanofibers with a diameter around of 1200 nm form a crosslink network structure. After surface modification treatment (PDA/PEI-coating) the color of the pristine PCL electrospun membranes changed from white to dark grey, and some randomly distributed small bumps could be observed on the surface of the modified PCL electrospun membranes as depicted in Figure 2b. When g-C_3_N_4_/CQDs composites were dissolved in the PCL electrospun solution, the synthesized PCL-g-C_3_N_4_/CQDs electrospun nanofibers could still retain crosslink network structure, and a specific amount of embedded g-C_3_N_4_/CQDs could be observed in Figure 2c. Similar to the pristine PCL electrospun membranes after surface modification treatment, small bumps which adhere to the surface of PCL nanofibers and exposed g-C_3_N_4_/CQDs composites also could be observed in the modified PCL-g-C_3_N_4_/CQDs electrospun membranes as shown in Figure 2d. Compared with PCL-g-C_3_N_4_/CQDs electrospun membranes, the surface modification by PDA/PEI was conducive for the adsorption of AFB_1_ to the vicinity of photodegradation sites and promoting the contact probability of AFB_1_ and g-C_3_N_4_/CQDs composites, which improves the photocatalytic efficiency.

### 3.2. Micro-Structure of g-C_3_N_4_/CQDs Composites

Figure 3 showed the TEM and HRTEM images of g-C_3_N_4_/CQDs composites, revealing the detailed microstructure. Apparently, many CQDs with diameters of 10–20 nm were uniformly decorated on g-C_3_N_4_ with numerous stacked block structures, which was consistent with the findings from previous studies [40,41]. In addition, a sharp dividing line in the HRTEM image further exhibited the coexistence of g-C_3_N_4_ and CQDs nanoparticles, which confirmed that CQDs homogeneously combined with g-C_3_N_4_. The micro-regional heterostructures between CQDs and g-C_3_N_4_ could effectively enhance different orientation intrinsic driving forces to accelerate the separation and transfer of photogenerated charges [42].

### 3.3. UV-vis Absorption Spectra and PL Spectra of g-C_3_N_4_ with Different Content of CQDs

To analyze the light capture ability of g-C_3_N_4_/CQD composites, UV-vis absorption spectra of g-C_3_N_4_/CQDs composites with different contents of CQDs (0.1%, 0.3%, 0.5%, and 0.7%) and pristine g-C_3_N_4_ were recorded, as shown in Figure 4a. Compared with pristine g-C_3_N_4_, the light absorption region of g-C_3_N_4_/CQDs was greatly enhanced with the increasing content of CQDs. Redshift was observed from the absorption edge of g-C_3_N_4_ at 463 nm, which may be due to the effective combination of CQDs into g-C_3_N_4_ by thermal polymerization. We performed photoluminescence (PL) analysis, the results were shown in Figure 4b, to study the recombination of photogenerated charges of g-C_3_N_4_/CQDs composites using an excitation wavelength of 360 nm. Both pristine g-C_3_N_4_ and g-C_3_N_4_/CQDs samples with different contents of CQDs (0.1%, 0.3%, 0.5%, and 0.7%) had emission peaks at about 460 nm originated from inter-band recombination of photogenerated charges. The implanted CQDs acted as electron-accepting and transport centers that could facilitate the photogenerated charges transfer greatly and promote π–electron delocalization in the micro-region, inhibiting the recombination of photogenerated charges. Thereby, the emission peaks dramatically decreased with the increase in CQDs content (0~0.5%) and the photocatalytic efficiency was improved accordingly [38]. However, when the content of CQDs raised to 0.7%, the PL emission intensity turned over. In this situation, excess CQDs not only weakened the light absorption of g-C_3_N_4_, but also acted as the recombination sites of photogenerated charges, leading to an increase in emission intensity and a decrease in photocatalytic efficiency [40].

### 3.4. XRD Analysis of the Electrospun Membranes and g-C_3_N_4_/CQDs Composites

Figure 5a showed the XRD patterns of pristine PCL electrospun membranes, modified PCL electrospun membranes, PCL-g-C_3_N_4_/CQDs electrospun membranes and modified PCL-g-C_3_N_4_/CQDs electrospun membranes. The typical diffraction peaks at 21.5° and 23.9° correspond to (110) and (200) crystal planes of PCL, respectively [28]. After surface modification treatment, two accompanying peaks at 22.2° and 24.5° appeared on the XRD patterns of modified PCL electrospun membranes and modified PCL-g-C_3_N_4_/CQDs electrospun membranes, confirming the structural change caused by the functionalization reaction [43]. This also revealed that the structure of PCL has not been destroyed by surface modification treatment as the two diffraction peaks of PCL stay the same in the four XRD patterns. As stated above, XRD patterns showed that those two electrospun membranes were successfully modified. Moreover, no diffraction peaks for g-C_3_N_4_/CQDs were observed due to the low content. The details of XRD patterns of four membranes from 15° to 30° can be seen in Appendix A. In order to analyze the crystal structure of the synthesized g-C_3_N_4_/CQDs composites, the XRD patterns of g-C_3_N_4_, CQDs and g-C_3_N_4_/CQDs composites were also obtained, as shown in Figure 5b. The characteristic peak of g-C_3_N_4_ appears at 2θ = 13.1° was assigned to the (100) plane, which was attributed to the triazine unit. Additionally, another stronger peak that appears at 27.6° was the typical (002) diffraction plane attributed to the inter-planar stacking of the aromatic system in g-C_3_N_4_ [44]. The XRD pattern of the CQDs exhibited a wide bump around 10°, corresponding to the (002) plane [45]. It is worth noting that due to the low content and uniform distribution of the CQDs, almost no diffraction peaks of CQDs were observed in the g-C_3_N_4_/CQDs composite, revealing that the incorporation of CQDs did not significantly change the crystal structure of g-C_3_N_4_. More details can be found in Appendix A.

### 3.5. XPS Analysis of the Modified Electrospun Membranes

X-ray photoelectron spectroscopy (XPS) was used to investigate the chemical state and composition of the modified PCL-g-C_3_N_4_/CQDs electrospun membranes as shown in Appendix A, which showed that it was completely composed of C, N and O elements. As the content of g-C_3_N_4_/CQDs in those electrospun membranes was low and played a decisive role in the photodegradation of AFB_1_, we further carried out XPS analysis on g-C_3_N_4_/CQDs composites. The XPS wide scan spectra of g-C_3_N_4_/CQDs was shown in Figure 6a, revealing the existence of C, N and O elements at binding energies of 288 (C 1s), 400 (N 1s) and 532 eV (O 1s). Further, we used the peak-splitting simulation method to understand the chemical properties. In the high-resolution XPS spectra of C 1s (Figure 6b), four peaks located at 285.1, 286.0, 288.7 and 294.1 eV were ascribed to graphitic carbon (C=C) of CQDs, a small quantity of C-O species, sp^2^ hybridized carbon (N-C=N) of g-C_3_N_4_ and π-excitation, respectively [42,46]. The spectra of N 1s (Figure 6c) could be fitted into four peaks at 398.9, 399.8, 401.2 and 404.5 eV, which could be assigned to sp^2^ hybridized aromatic N (C-N=C) in triazine rings, tertiary N in N-(C)_3_ and amino groups (N-H), π-excitation, respectively [40,47]. The spectra of O 1s (Figure 6d) could be fitted into two peaks at 531.7 and 533.4 eV, which could be assigned to C-O and O-H of lattice oxygen and adsorbed water, respectively [42,46]. In the whole spectrum, no characteristic peak related to any other element was observed both in modified PCL-g-C_3_N_4_/CQDs electrospun membranes and g-C_3_N_4_/CQDs composites indicating this metal-free material can avoid the release of poisonous metal ions during practical application.

### 3.6. FTIR Analysis of the Electrospun Membranes and g-C_3_N_4_/CQDs Composites

Fourier transform infrared spectra (FTIR) can be used to characterize electrospun membranes and g-C_3_N_4_/CQDs composites. Figure 7a showed the FTIR spectra of the four membranes, which were almost the same. The PCL-related absorption peaks were observed at 2949 cm^−1^ (asymmetric -CH_2_- stretching), 2868 cm^−1^ (symmetric -CH_2_- stretching), 1727 cm^−1^ (-C=O carbonyl stretching), 1293 cm^−1^ (C–O and C–C stretching in the crystalline phase), 1240 cm^−1^ (asymmetric C–O stretching), 1190 cm^−1^ (symmetric C–O stretching), and 1157 cm^−1^ (C–O and C–C stretching in the crystalline phase) [48,49]. Whether the electrospun membranes were surface-modified or anchored with g-C_3_N_4_/CQDs composites, their FTIR results were basically the same. The reason may be that the PDA/PEI molecules and g-C_3_N_4_/CQDs composites were too few to detect. The same was true of the work of other researchers, such as Scaffaro et al. [49], and Xu et al. [50]. Figure 7b showed the FTIR spectrums of the g-C_3_N_4_ and g-C_3_N_4_/CQDs composites. The main absorption peaks at 3200 cm^−1^ (N–H stretching), 1637 cm^−1^ (C=N stretching), 1574 cm^−1^ (C=N stretching), 1405 cm^−1^ (C-N stretching), 1316 cm^−1^ (C-N stretching), 1236 cm^−1^ (C-N stretching), and 810 cm^−1^ (vibration mode of 3-*s*-triazine unit) belonged to g-C_3_N_4_ [40,47]. It was clear that the g-C_3_N_4_ and g-C_3_N_4_/CQDs composites had almost the same FTIR spectra, indicating that the introduction of CQDs did not significantly change the chemical structure of g-C_3_N_4_.

### 3.7. Performance of Removing AFB_1_

The adsorption-removal performances of AFB_1_ aqueous solution were evaluated in dark and the adsorption experiments were 30 min. We could see from Figure 8a that the concentrations of AFB_1_ were reduced to 87.1% and 84.2% with pristine PCL electrospun membranes and PCL-g-C_3_N_4_/CQDs electrospun membranes immersed in 30 min, respectively. However, the adsorption-removal efficiencies were significantly improved and up to 63.5% and 58.6% when using modified PCL electrospun membranes and modified PCL-g-C_3_N_4_/CQDs electrospun membranes. It could be inferred that the surface modification by PDA/PEI coating not only improved the hydrophilicity of the membranes (Appendix A) but also endowed the membranes with extremely strong adsorption capacity for AFB_1_. Moreover, we saw that the hydrophobic interaction between the modified membranes and AFB_1_ was a key factor for the adsorption mechanism. The HPLC chromatogram of AFB_1_ aqueous solution concentrations over the adsorption time with modified PCL-g-C_3_N_4_/CQDs electrospun membranes was demonstrated in Figure 8b. When the experiments were carried out under visible light irradiation, the concentrations of AFB_1_ decreased rapidly with PCL-g-C_3_N_4_/CQDs electrospun membranes and modified PCL-g-C_3_N_4_/CQDs electrospun membranes immersed. After 30 min of visible light radiation, only 16.1% and 3.1% of AFB_1_ were left over, as shown in Figure 8c. Moreover, the removal efficiencies of the other two membranes without g-C_3_N_4_/CQDs remained almost unchanged compared with the adsorption experiments. These results unambiguously showed that the removal of AFB_1_ under visible light irradiation was due to synergistic adsorption and photocatalytic degradation. The HPLC chromatogram of AFB_1_ aqueous solution concentrations over the irradiation time with modified PCL-g-C_3_N_4_/CQDs electrospun membranes was demonstrated in Figure 8d. Different from adsorption experiments, intermediate products would be produced during the process of photocatalysis, depending on the type of photocatalysts and reaction conditions [51]. The chromatogram peaks appearing at 8.5–9.5 min in Figure 8d were attributed to intermediate products [51], and most of them were eliminated by adsorption and photocatalysis.

To understand more about the adsorption behavior, the pseudo-first-order [Equation (1)] and pseudo-second-order [Equation (2)] kinetic models were then used to analyze the adsorption kinetics process [12,52].
log(*q_e_* − *q_t_*) = log*q_e_* − *k_1_t*/2.303(1)
*t*/*q_t_* = 1/*k_2_q_e_*^2^ + *t*/*q_e_*(2)
where *q_e_* and *q_t_* (μg/mg) represent the equilibrium adsorption amount and the adsorption amount at time *t*; *k_1_* (1/min) and *k_2_* (mg/(μg·min)) are the rate constants for the first- and second-order adsorption process, respectively. The fitting results were illustrated in Figure 9. The relevant kinetic parameters calculated from the fitting curves were listed in Table 1. In general, the pseudo-first-order model assumes that the adsorption process is mainly due to physical adsorption, whereas the pseudo-second-order model indicates that the whole adsorption process is dominated by chemical adsorption. Both instances of *R^2^* in the pseudo-first-order model (0.99236) and pseudo-second-order model (0.98694) were greater than 0.95, indicating that physical adsorption and chemical adsorption both contributed to the removal of AFB_1_ and exhibited mainly physical adsorption accompanied by chemical adsorption [53].

To test the stability of the modified PCL-g-C_3_N_4_/CQDs electrospun membranes, we conducted 5 consecutive experiments under the same experimental conditions. Figure 10 showed the reproducibility results of AFB_1_ degradation. We can learn that the degradation rate reached more than 96% within 30 min after five consecutive experiments. The recyclability of the modified PCL-g-C_3_N_4_/CQDs electrospun membranes shows the possibility of its practical application as well as better economic benefits.

### 3.8. Mechanism for Enhanced Degradation Performance

To better understand the mechanism of synergistic adsorption and photocatalytic degradation for AFB_1_ by the modified PCL-g-C_3_N_4_/CQDs electrospun membranes, modified PCL electrospun membranes and g-C_3_N_4_/CQDs composites were used for adsorption and photocatalysis experiments, respectively, under the same conditions described above. NaCl and urea were added into AFB_1_ solutions before adsorption experiments, and the AFB_1_ removal rate decreased to 41.3% and 7.31%, as shown in Figure 11a. NaCl and urea are well-known “killers” of electrostatic and hydrogen bonds [54], which indicated that electrostatic attraction and hydrogen bonds were the main adsorption mechanisms between AFB_1_ molecules and PDA/PEI coating. For the photocatalysis experiments, AO, IPA and BQ were employed as the scavengers for h^+^, •OH and•O_2_^−^, respectively [55]. Figure 11b displays that the degradation rate of the AFB_1_ solution without a sacrificial agent was 97.2% after 5 min of visible light irradiation, whereas those with scavengers such as IPA, BQ, and AO were 79.6%, 4.6%, and 96.7%, respectively. Based on the above results, we learned that both •O_2_^−^ and •OH contributed to AFB_1_ degradation, and •O_2_^−^ played the dominant role.

The possible adsorption and photocatalytic mechanism of AFB_1_ degradation by the modified PCL-g-C_3_N_4_/CQDs electrospun membranes was proposed, as shown in Figure 12. PDA/PEI coating on the surface of the membranes adsorbs and intercepts AFB_1_ molecules depending on electrostatic attraction and hydrogen bonds firstly. Upon visible light irradiation, photogenerated electron-hole pairs are generated in the g-C_3_N_4_ component of the g-C_3_N_4_/CQDs composites on the membranes. The generated h^+^ stayed in the valence band of g-C_3_N_4_, whereas e^-^ migrates to the conduction band. As the valence band of g-C_3_N_4_ is more negative than the potential of E(OH^−^/·OH) or E(H_2_O/·OH) (1.53V < 1.99 or 2.4V) [40], the h^+^ would not react with H_2_O/OH^−^ to produce ·OH. Moreover, the implanted CQDs act as electron-accepting and transport centers enhance the separation of photogenerated electron-hole pairs and promote the generation of •O_2_^−^ [56]. Part of •O_2_^−^ oxidizes H_2_O to •OH [40]. At last, the •O_2_^−^ and •OH can decompose AFB_1_ molecules into smaller fragments, even CO_2_ and H_2_O. After the reaction, the modified PCL-g-C_3_N_4_/CQDs electrospun membranes are regenerated as the initial, which can be immediately used in a new process without any further treatment.

## 4. Conclusions

In summary, we employed electrospinning and surface modification technology to prepare novel modified PCL-g-C_3_N_4_/CQDs electrospun membranes. It was observed that adsorption significantly accelerated the process of photodegradation according to contrast experiments. More importantly, the as-prepared membranes can continuously eliminate AFB_1_ through the synergistic effects of adsorption and photocatalysis; moreover, regeneration is a green approach synchronized with the reaction under visible light without any physical or chemical treatment. The adsorption mechanism in which electrostatic attraction and hydrogen bonds play major roles was revealed, and the photodegradation mechanism of AFB_1_ using g-C_3_N_4_/CQDs composites was studied based on active species trapping experiments. The reusability and stable activity were confirmed during five cycles of degradation experiments. Thus, the novel modified PCL-g-C_3_N_4_/CQDs electrospun membranes demonstrated easy separation, good reusability and provided a new insight into the design of high-performance membranes for the degradation of AFB_1_ in practical application.

## Figures and Tables

**Figure 1 biomolecules-13-00550-f001:**
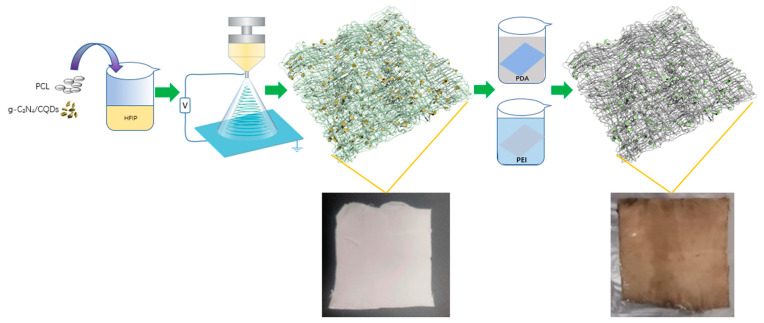
The schematic illustration of the fabrication of modified PCL-g-C_3_N_4_/ CQDs electrospun membranes.

**Figure 2 biomolecules-13-00550-f002:**
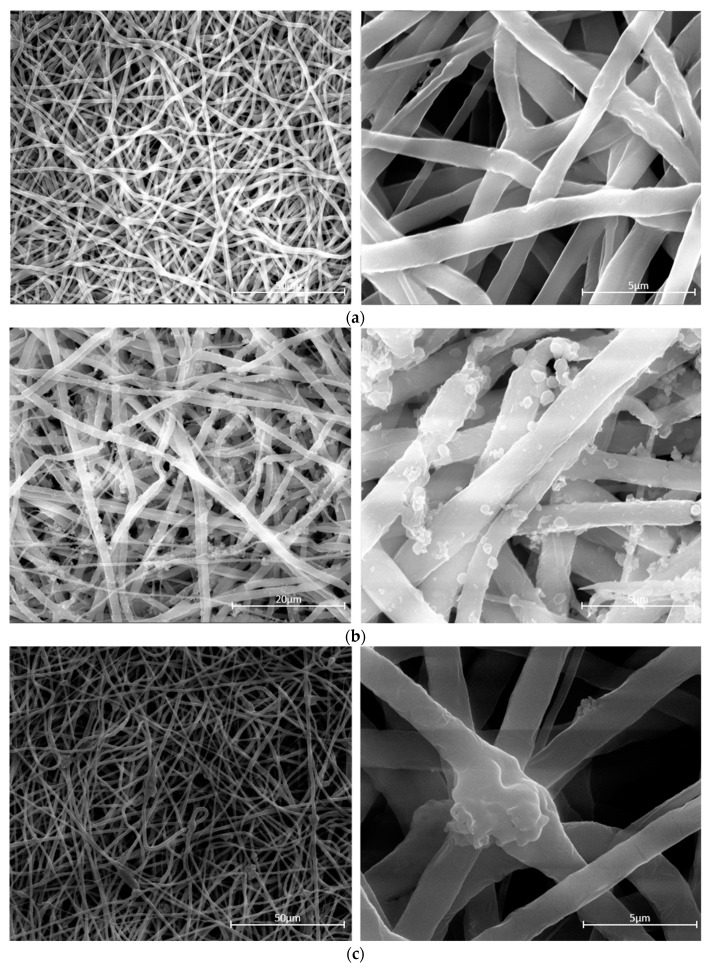
SEM images of (**a**) pristine PCL electrospun membranes, (**b**) modified PCL electrospun membranes, (**c**) PCL-g-C_3_N_4_/CQDs electrospun membranes, and (**d**) modified PCL-g-C_3_N_4_/CQDs electrospun membranes.

**Figure 3 biomolecules-13-00550-f003:**
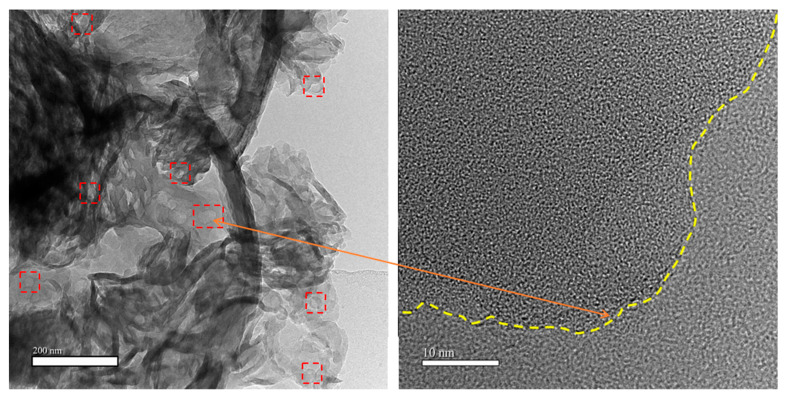
TEM and HRTEM images of g-C_3_N_4_/CQD composites.

**Figure 4 biomolecules-13-00550-f004:**
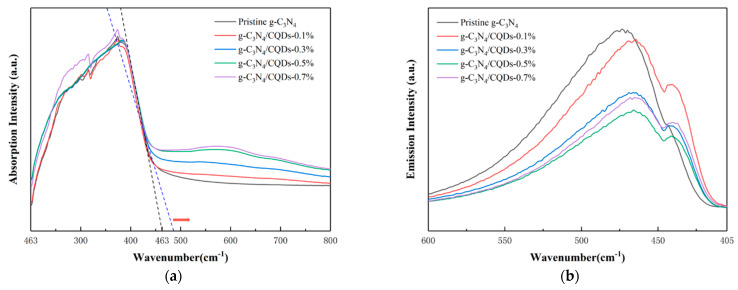
(**a**) UV-vis absorption spectra and (**b**) PL spectra of g-C_3_N_4_ with different content of CQDs from 0~0.7%.

**Figure 5 biomolecules-13-00550-f005:**
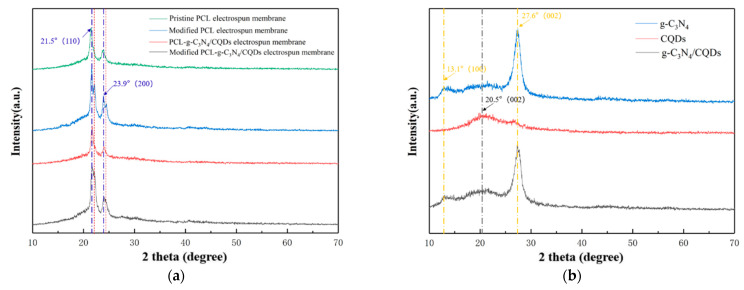
XRD patterns of (**a**) four membranes and (**b**) g-C_3_N_4_, CQDs, g-C_3_N_4_/CQDs.

**Figure 6 biomolecules-13-00550-f006:**
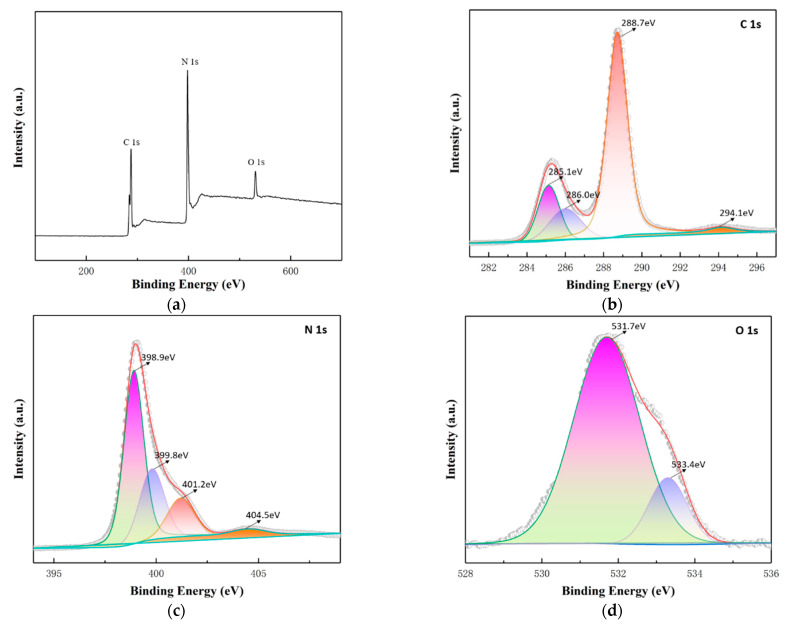
XPS survey spectrum of g-C_3_N_4_/CQDs composites: (**a**) the full-scale XPS spectrum, high-resolution XPS spectra of (**b**) C 1s, (**c**) N 1s, (**d**) O 1s.

**Figure 7 biomolecules-13-00550-f007:**
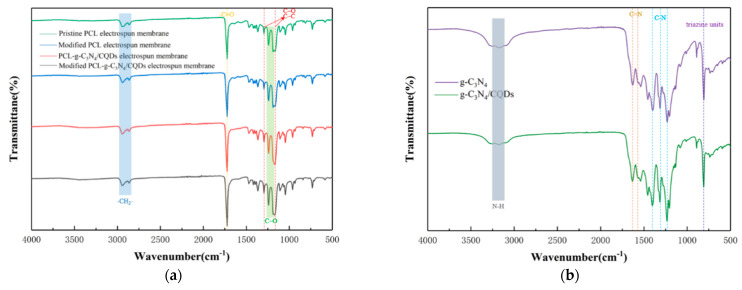
FTIR spectra of (**a**) the as-prepared membranes and (**b**) g-C_3_N_4_, g-C_3_N_4_/CQDs composites.

**Figure 8 biomolecules-13-00550-f008:**
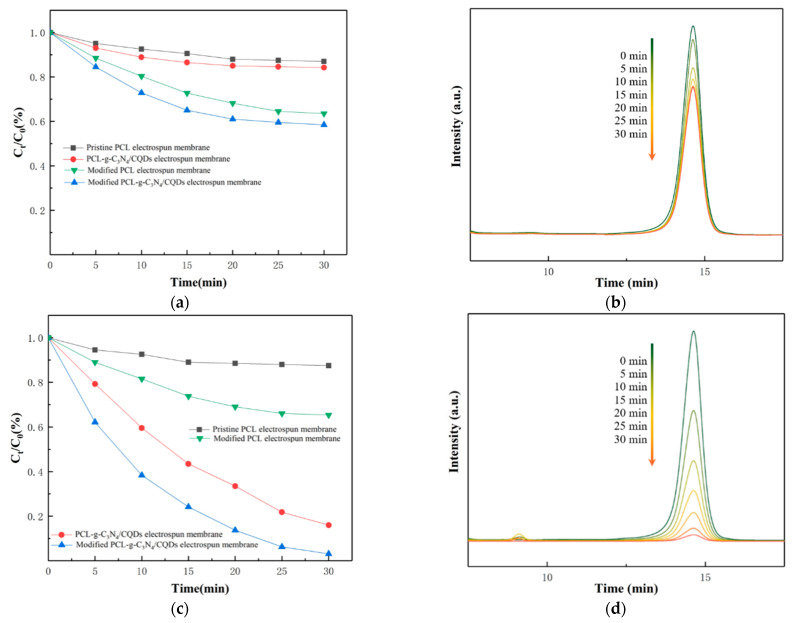
(**a**) Adsorption-removal performances of AFB_1_ with the four kinds of membranes under dark. (**b**) HPLC chromatogram of AFB_1_ adsorption-removal with modified PCL-g-C_3_N_4_/CQDs electrospun membranes under dark at different times. (**c**) Synergistic adsorption and photocatalytic degradation performance of AFB_1_ with the four kinds of membranes under visible light irradiation. (**d**) HPLC chromatogram of AFB_1_ adsorption and photocatalytic degradation with modified PCL-g-C_3_N_4_/CQDs electrospun membranes under visible light irradiation at different times.

**Figure 9 biomolecules-13-00550-f009:**
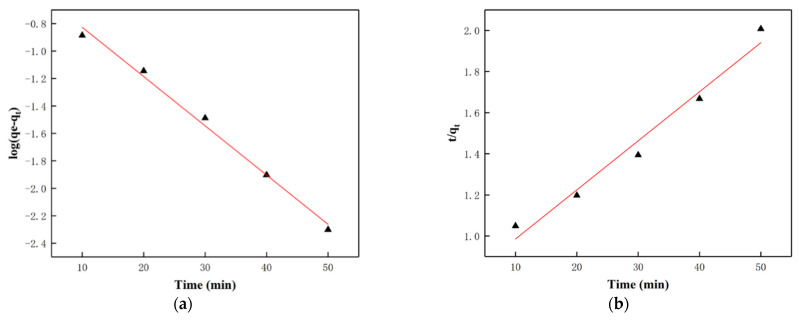
Corresponding linear fitting of AFB_1_ adsorption by the modified PCL-g-C_3_N_4_/CQDs electrospun membranes with time, (**a**) pseudo-first-order model and (**b**) pseudo-second-order model.

**Figure 10 biomolecules-13-00550-f010:**
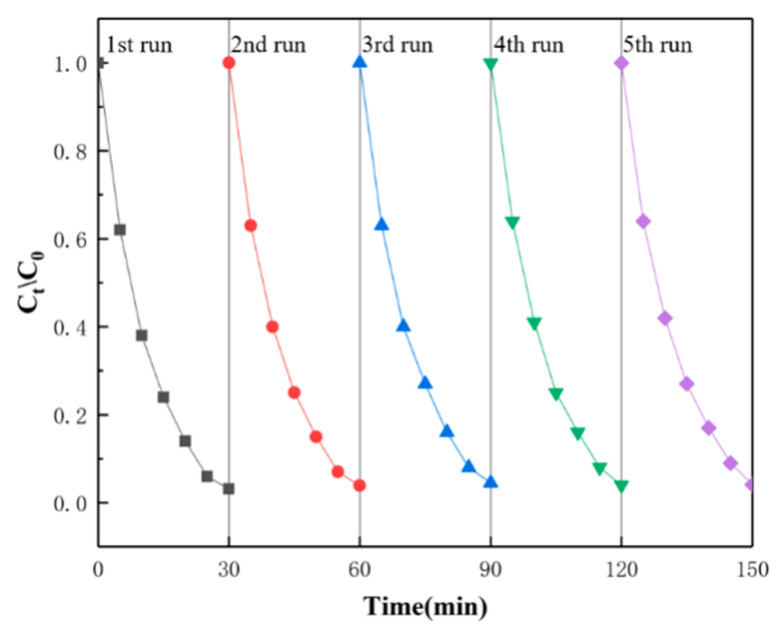
The reproducibility of the modified PCL-g-C_3_N_4_/CQDs electrospun membranes for degradation of AFB_1_ for 5 cycles.

**Figure 11 biomolecules-13-00550-f011:**
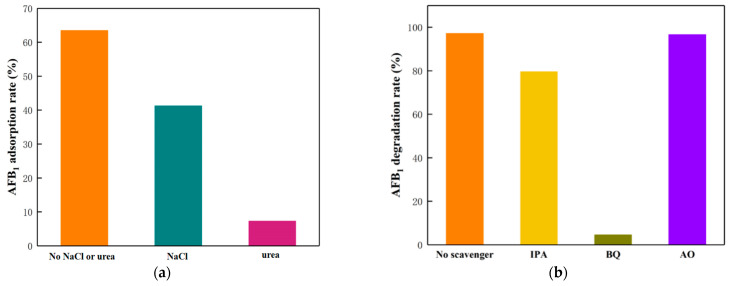
(**a**) Adsorption activities of modified PCL electrospun membranes in the presence of NaCl/urea (or not). (**b**) Photocatalytic activities of g-C_3_N_4_/CQDs composites for the degradation of AFB_1_ in the presence of different scavengers.

**Figure 12 biomolecules-13-00550-f012:**
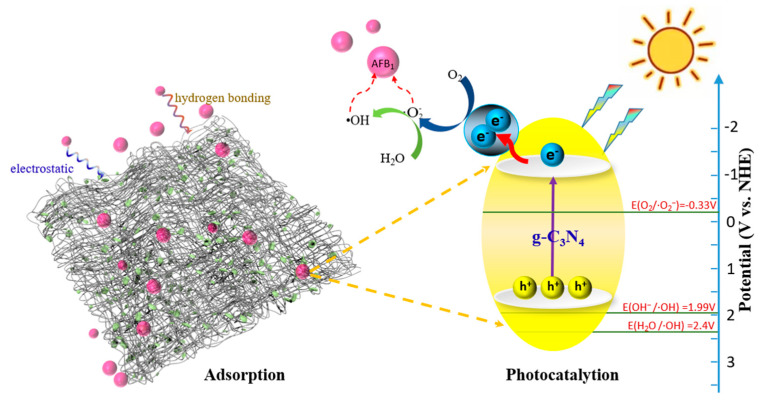
The adsorption and photocatalytic mechanism of removal/degradation of AFB_1_ for the modified PCL-g-C_3_N_4_/CQDs electrospun membranes.

**Table 1 biomolecules-13-00550-t001:** Kinetics parameters for AFB_1_ adsorption removal.

Sample	*q_e_* (μg/mg)	Pseudo-First Order		Pseudo-Second Order	
		log(*q_e_ − q_t_*) = log*q_e_ − k_1_t*/2.303		*t/q_t_ =* 1/*k_2_q_e_^2^ + t/q_e_*	
		*k_1_* (1/min)	*R^2^*	*k_2_* (mg/(μg·min))	*R^2^*
Modified PCL-g-C_3_N_4_/CQDs electrospun membranes	0.2075	0.03589	0.99236	0.02378	0.98694

## Data Availability

Raw data are available on request.

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
