# Peer review of "Enhancement of AFB1 Removal Efficiency via Adsorption/Photocatalysis Synergy Using Surface-Modified Electrospun PCL-g-C3N4/CQDs Membranes"

_biomolecules, 2023, doi:10.3390/biom13030550_

Round 1

Reviewer 1 Report

The manuscript prepared a novel modified PCL-g-C3N4/CQDs electrospun membranes employing electrospinning and surface modification technology, which can continuously eliminate AFB1 by the synergistic effect of adsorption and photocatalysis under visible light. The study designs well and the results provide very interesting information. However, some comments for the manuscript need to be addressed before it can be accepted.

1. Line 24: the word “photocatalytic” should be changed to the noun form “photocatalysis”.

2. Lines 53-56: Except for adsorption and photocatalysis used in this study, it is necessary to add brief introduction of other common detoxification methods of AFB1, such as animal function regulator (curcumin), biodegradation (probiotics) as the part of this paragraph, and allow me to suggest the following publications to be cited herein, please.

Wang, Y.; Liu, F.; Zhou, X.; Liu, M.; Zang, H.; Liu, X.; Shan, A.; Feng, X. Alleviation of Oral Exposure to Aflatoxin B1-Induced Renal Dysfunction, Oxidative Stress, and Cell Apoptosis in Mice Kidney by Curcumin. Antioxidants 2022, 11, 1082. https://doi.org/10.3390/antiox11061082.

3. Lines 145-146: AFB1 was dissolved with deionized water. Are the authors sure about the toxin dissolved in water, since AFB1 is very slightly soluble in water?

4. Line 156: the word “compo$sites” should be “composites”.

5. The section of “3.1 Characterization of the PAN-g-C3N4/ CQDs Electrospun Membranes” is too long and the logic is not very clear. It will be more logical to divide this section into several sub-sections, such as 3.1 Micro-structure of PAN-g-C3N4/ CQDs Electrospun Membranes, 3.2 UV–vis Absorption Spectra of g-C3N4 with different content of CQDs, etc.

6. Figure 11(a), the legend of colums: “Nacl” should be “NaCl”

7. Line 475, provide the full name of the abbreviation “RhB”, “Rhodamine B”?

Author Response

Dear Professor:
Thank you for your comments concerning our manuscript entitled “Enhancement of AFB1 removal efficiency via adsorption/photocatalysis synergy using surface modified electrospun PCL-g-C3N4/CQDs membranes” (ID: ISSN 2218-273X). Those comments are all valuable and very helpful for revising and improving our paper, as well as the important guiding significance to our research. We have studied the comments carefully and made corrections which we hope meet with approval. Revised portions are marked in red in the paper. The responses to your comments please see the attachment. Thank you!

Reviewer 2 Report

This work reports on the photodegradation of aflatoxin B1 (AFB1) residues by using a modified polycaprolactone membrane that loaded with g-C3N4/CQDs. The authors have considered the main factors required to degrade AFB1 residues by photocatalysis. I see the experiments are carefully designed, together with the systematic characterizations, evaluation of performance, and analysis of mechanism. Data are informative. Basically, I see no flaw in this work. It should be published in the Journal of Biomolecules after some minor revisions are made.

1. Do we need to worry about the product of AFB1 after photodegradation? Is it desorbed in solution or immobilized on the modified fibers?Is there a loading capacity for this kind of electrospun PCL-g-C3N4/CQDs membrane?

2. A minor typo “$” in “2.2 Preparation of g-C3N4/CQDs compo$sites”.

Author Response

(The authors gave the same response as above.)

Reviewer 3 Report

In manuscript Enhancement of AFB1 removal efficiency via adsorption/photocatalysis synergy using surface modified electrospun PCL-g-C3N4/CQDs membranes authors prepared membranes for removing AFB1 from matrix and proposed to use these membrane sin practice.

In introduction part the most important known facts on AFB1 toxicity, its incidence and regulation are written. Authors also explained importance of control growth of the Aspergillus moulds but also importance of detoxification. Good overview of known adsorption materials and methods are presented. Authors also presented advantages electrospinning adsorption membranes and their possible problems or unknown facts they want to investigate and are these membranes environmentally friendly. Mechanism of photocatalytic degradation and importance of using this technology is well explained. Aim of the study is clear. From the introduction part it seems that authors studied literature thoroughly.

Preparation of composites (there is a typo in 2.2. compos$ites) and membranes, their characterization and photocatalytic degradation are explained in detail and well written.

Results are clearly presented accompanied with a lot of figures and compared and discussed with the known facts from the literature. Green approach, which is very important, is well presented.

Conclusion is based on experiment results. Used literature is appropriate and recently published.

Overall, the manuscript is very well written, methodologically and linguistically. It is easy to read and understand because it is inovative, interesting and everything is well explained.

Author Response

(The authors gave the same response as above.)

Reviewer 4 Report

The paper entitled "Enhancement of AFB1 removal efficiency via adsorption/photocatalysis synergy using surface modified electrospun PCL-g-C3N4/CQDs membranes” by Yao L. et al. has been reviewed. The paper deals with the important problem of removal of aflatoxin B1. The experimental results obtained for the membranes with different composition and surface treatment seem to be very promising. I recommend the publication of this work, but I do have some remarks as to the contents of the manuscript. 1.       To begin with, there are mistakes in the English language (for example, line 19; line 24; line 42; line 133; line 135; line 156; line 228; lines 248-251; line 280; line 317; line 325; line 34; line 353; line 360; lines 379-318; lines 433-435; line 445, etc.). 2.       In title of sections 2.3 and 3.1 as well as in Caption to Fig. 1 PAN is mentioned. Does it mean PCL? 3.       More details of FTIR and XPS experiments should be added (resolution, mode, type of XPS spectrometer and eventual calibration). 4.       The zoom of 5-30° zone should be given for clarity in Fig. 5. 5.       The values of contact angle measurements should be added (in addition to Fig. S3). 6.       The measurements precision should be given in Fig. 8, 9 and 11. 7.       Why the membranes were tested only during 5 cycles (Fig. 10)? For practical application, it would be interesting to see till which time the membrane will stay efficient. 8.       The authors mentioned “crosslink network structure” for the membranes obtained by electrospinning. The nature of crosslinking (physical or chemical) should be detailed.

Author Response

(The authors gave the same response as above.)

Round 2

Reviewer 1 Report

All my comments have been addressed in the revised manuscript, I consider that it is suitable for publication in the journal of Biomolecules.